# Non-invasive prenatal testing (NIPT) by low coverage genomic sequencing: Detection limits of screened chromosomal microdeletions

**Marcel Kucharik[1,2], Andrej Gnip[3,4], Michaela Hyblova[3,4], Jaroslav Budis[1,2,5], Lucia Strieskova[1], Maria Harsanyova[1,6], Ondrej Pös[1,6], Zuzana Kubiritova[1,6,7] *, Jan Radvanszky[1,7], Gabriel Minarik[3,4], Tomas Szemes[1,2,6]**

1 Geneton Ltd., Bratislava, Slovakia, 2 Comenius University Science Park, Bratislava, Slovakia, 3 Medirex Inc., Bratislava, Slovakia, 4 TrisomyTest Ltd., Bratislava, Slovakia, 5 Slovak Centre of Scientific and Technical Information, Bratislava, Slovakia, 6 Department of Molecular Biology, Faculty of Natural Sciences, Comenius University, Bratislava, Slovakia, 7 Institute of Clinical and Translational Research, Biomedical Research Center, Slovak Academy of Sciences, Bratislava, Slovakia

* zuzana.kubiritova@savba.sk

**Data Availability Statement:** Data used for training and testing are available from GitHub (https://github.com/marcelTBI/CNV_data). These data

## Abstract

To study the detection limits of chromosomal microaberrations in non-invasive prenatal testing with aim for five target microdeletion syndromes, including DiGeorge, Prader-Willi/Angelman, 1p36, Cri-Du-Chat, and Wolf-Hirschhorn syndromes. We used known cases of pathogenic deletions from ISCA database to specifically define regions critical for the target syndromes. Our approach to detect microdeletions, from whole genome sequencing data, is based on sample normalization and read counting for individual bins. We performed both an *in-silico* study using artificially created data sets and a laboratory test on mixed DNA samples, with known microdeletions, to assess the sensitivity of prediction for varying fetal fractions, deletion lengths, and sequencing read counts. The *in-silico* study showed sensitivity of 79.3% for 10% fetal fraction with 20M read count, which further increased to 98.4% if we searched only for deletions longer than 3Mb. The test on laboratory-prepared mixed samples was in agreement with *in-silico* results, while we were able to correctly detect 24 out of 29 control samples. Our results suggest that it is possible to incorporate microaberration detection into basic NIPT as part of the offered screening/diagnostics procedure, however, accuracy and reliability depends on several specific factors.

## Introduction

In recent years, prenatal care has been attempting to determine the genetic background of a fetus with a decreasing risk of miscarriage. One of the most important milestones in this field was the discovery of cell-free fetal DNA (cffDNA) in the plasma of pregnant women [1]. It has led to the emergence of a new basic and applied research field of non-invasive prenatal testing (NIPT) [2]. Different commercial companies, e.g. Sequence, BGI, Illumina, Natera, Roche,

include anonymized mapped data without genomic information to ensure participant confidentiality. The repository contains all needed data and scripts for reproduction of results of the article. The scripts and the data are for non-commercial use only, since they are part of a commercially used tests Trisomy Test + and Trisomy Test Complete (https://trisomytest.sk/en/) and are intellectual properties of Geneton Ltd.

**Funding:** This work was supported by the "REVOGENE – Research centre for molecular genetics" project (ITMS 26240220067) supported by the Operational Programme Research and Development funded by the European Research and Development Fund (ERDF) (https://ec.europa.eu/regional_policy/en/funding/erdf/) and APVV project- Slovak Research and Development Agency project (https://www.apvv.sk/?lang=en) with number APVV-15- 0232. The funders Geneton Ltd., Medirex Inc., Trisomy Test Ltd. provided support in the form of salaries for authors MK, JB, LS, MaH, OP, ZK, JR, TS, AG, MiH, GM, respectively, but did not have any additional role in the study design, data collection and analysis, decision to publish, or preparation of the manuscript. The specific roles of these authors are articulated in the 'author contributions' section.

**Competing interests:** I have read the journal's policy and the authors of this manuscript have the following competing interests: We declare potential competing financial interest in the form of employee contracts (see affiliations for each author) with Geneton Ltd. and TrisomyTest Ltd.. Geneton Ltd. participated in the development of a commercial NIPT test in Slovakia, however is not a provider of this commercial test, but still continues to do basic and applied research in the field of NIPT. On the other hand, TrisomyTest Ltd. is the commercial providers of NIPT testing in Slovakia, their participation in the study was, however, limited to the routine NIPT testing that generated the genomic results reused in our study. Related to this work, there are no patents, products in development or marketed products to declare. This does not alter our adherence to PLOS ONE policies on sharing data and materials, however, there are some restrictions in sharing of our data publicly. More details on these can be found in the Data Availability Statement.

LifeCodexx, are utilizing results of this research in their services focused on NIPT of T13, T18, T21, and sex chromosome aneuploidies. Furthermore, they are broadening their product portfolio with tests for detection of most common chromosomal microdeletions. Such tests can be used to identify pregnancies with high risk of DiGeorge syndrome, 1p36 syndrome, Cri-Du-Chat syndrome, Prader-Willi/Angelman syndrome, and Wolf-Hirschhorn syndrome. However, validation of methods for detection of microdeletions is very limited. While sensitivity and specificity of detection of most common trisomies and sex chromosomes aneuploidies were published in large meta-analyses [3–5], these important test parameters for detection of the above mentioned microdeletions are very scarce [6–8]. Furthermore, the sensitivity for different input parameters such as fetal fraction and aberration size is not evaluated, thus the importance of each parameter is not quantified. This is due to very low prevalence of such syndromes, resulting in very limited clinical validation data sets. Therefore, currently available proof of principle studies focused on chromosomal microdeletions detection and corresponding validation studies through analyses of artificial data. These included samples prepared either by *in-silico* massively parallel sequencing data manipulation, or by mixing-up of normal and well defined microdeletion positive DNA samples with subsequent testing on few real clinical samples [6,9]. Microdeletion syndromes detection sensitivity and specificity, similarly to most common trisomies detection, was found to be the most dependent on technical and biological parameters of sample and the test itself. Coverage of target region, fetal fraction, size, and position of the deletions were identified as the most prominent factors [9].

Recent years yielded a handful of tools for CNV detection from (shallow) whole-genome sequencing data, some even with comparisons between available tools [10]. However, none of them have thoroughly studied the importance of technical and biological parameters (such as deepness of sequencing, fetal fraction, or variation length) to prediction accuracy. All the tools for CNV detection behave similarly in a way, that first they bin the reads and work only with bin counts, then employ some form of normalization and noise correction and lastly segment the normalized signal and call variations. The notable CNV detection tools include: Wisecondor X [10] (successor of the Wisecondor [11] tool), CNVkit [12], CNVnator [13], iCopyDav [14].

In present work, we tested detection limits for different sizes and positions of known microdeletions causing clinically relevant syndromes using an in-house CNV detection tool (available at https://github.com/marcelTBI/CNV_data). We identified syndrome specific critical regions ranging from 0.9 Mb to 21 Mb in pooled data from the public database of the International Standards for Cytogenomic Arrays (ISCA) Consortium [15] Subsequently, we estimate the sensitivity of detection in such artificially prepared data, mimicking the sizes and positions of pathogenic deletions. The experiment was split into two parts. The first analyses were performed on data prepared by artificially "spiking in" reads into sequencing data from physiological pregnancies. In the second experiment, analyses were performed in a blinded manner on artificially prepared sample mix-ups, with defined proportions of normal and well-defined microdeletions containing control DNA, mimicking different fetal fractions ranging from 5% to 20%.

## Material and methods

The study has been approved by the Ethical Committee of the Bratislava Self-Governing Region (Sabinovska ul.16, 820 05 Bratislava) on 30 April 2015 under the decision ID 03899_2015. In each relevant case written informed consents consistent with the Helsinki declaration were obtained.

## Retrieval of pathogenic regions

The ISCA database [15] was searched for deletions located in genomic regions associated with five selected microdeletion syndromes (22q11.2 –DiGeorge syndrome, 4p16.3 –Wolf-Hirschhorn syndrome, 15q11 –Angelman/Prader-Willi syndrome, 5p15 –Cri-Du-Chat syndrome, 1p36 – 1p36 deletion syndrome), using the hg19/GRCh37 version of the human reference genome. Coverage was calculated as the number of deletions which included the given nucleotide position, separately for pathogenic (or likely pathogenic) and benign (or likely benign) deletions.

Initially, coverage of 45 pathogenic deletions and higher was used as a cut-off to specify boundaries of the pathogenic regions. Afterwards, the regions were checked for outliers (pathogenic deletions located entirely outside the given region), and the coverage cut-off was gradually decreased until the pathogenic region was defined in such a way, that there were no outliers (in the case of Wolf-Hirschhorn, Angelman/Prader-Willi, and Cri-Du-Chat syndromes), or that the outliers were considered not relevant (in the case of DiGeorge and 1p36 deletion syndromes). The pathogenic region for Wolf-Hirschhorn syndrome defined by coverage cut-off 45 was the only one which did not have any outliers initially Fig 3. The region for DiGeorge syndrome defined by the same cut-off had a considerable number of outliers located towards the 3'-end, but these corresponded to another well described syndrome, known as 22q11.2 distal deletion syndrome [16]. Therefore, no further adjustment of the region boundaries were needed. In order to address the phenomenon of outliers, the coverage cut-off was reduced to 25 for Angelman/Prader-Willi syndrome, to 28 for Cri-Du-Chat syndrome, and to 3 for 1p36 deletion syndrome. There were still considerable numbers of overhangs from the defined pathogenic regions (pathogenic deletions spanning outside of the regions), but as long as there were no pathogenic deletions located entirely outside of them, it was assumed that the defined regions cover genes responsible for the syndromes. The two outliers found in the 1p36 genomic region (with positions chr1:27,133,503–28,011,702 and chr1:27,927,633–28,215,952 on hg19 human genome assembly) were considered a distinct entity from the deletions responsible for the 1p36 deletion syndrome and were thus excluded from the analysis.

Three pathogenic deletions in the 22q11.2 genomic region had overlap with DiGeorge syndrome critical region smaller or equal to only 20kb and in all three cases the overlap was located on the 3'-end of the critical region. These were assumed to be related to the 22q11.2 distal deletion syndrome and were excluded from further analysis. The final set of the microdeletions are summarized in Table 1. In total, we used 533 microdeletions for evaluation of our methods.

## Preparation of artificial NIPT data sets

Sequencing data from healthy NIPT samples can be used for identifying the limits of microaberration detection. Reads of such samples are binned into equal size bins according to read

**Table 1. Counts of microdeletions from the ISCA database.**

| Syndrome | Deletion size | | | | | | |
|---|---|---|---|---|---|---|---|
| | 0-1Mb | 1-2Mb | 2-3Mb | 3-4Mb | 4-5Mb | 5-10Mb | 10-40Mb |
| 1p36 deletion | 8 | 20 | 15 | 13 | 12 | 27 | 5 |
| Wolf-Hirschhorn | 0 | 5 | 5 | 12 | 5 | 11 | 10 |
| Cri-du-chat | 0 | 2 | 2 | 3 | 4 | 8 | 31 |
| Angelman/Prader-Willi | 6 | 0 | 0 | 1 | 48 | 34 | 1 |
| DiGeorge | 12 | 22 | 205 | 6 | 0 | 0 | 0 |

start. We used bin size of 20kb, although bin size of 50kb was reported to be used in previous study [9].

Artificial data sets mimicking aberrated samples were created from data sets belonging to healthy samples by multiplication of bins corresponding to the pathogenic regions. The multiplication coefficients were guided by a target fetal fraction. Specifically, for a target fetal fraction *ff*, all bins between the start and the end of the simulated pathogenic region were multiplied with *(1-ff/2)* for simulation of a chromosomal microdeletion or with *(1+ff/2)* for a chromosomal microduplication. Since it is known, that the fetal fraction is not constant throughout the whole genome [17], we apply the multiplication after both normalization procedures aimed to normalize the differences within a sample such as GC content bias and differences between different parts of the genome.

We created an artificially aberrated data set for each combination of NIPT data set and pathogenic region (identified in previous chapter) with either varying fetal fraction ratio in [0.05, 0.075, 0.1, 0.125, 0.15, 0.175, 0.2] or varying read count from 5M to 20M with a step of 1M. All artificial data sets were evaluated by the same algorithm described below.

## Identification of microaberrations

To identify microaberrations, we employed an approach similar to that described by Zhao et al. [9]. Briefly, we thoroughly normalized the bin counts and then used the circular binary segmentation algorithm on these bin counts to identify consistent segments of same coverage. The segments were then evaluated to determine their significance. More details are in later sections. Significant deviations are visualized for individual chromosomes (Fig 1) and for the target syndromes.

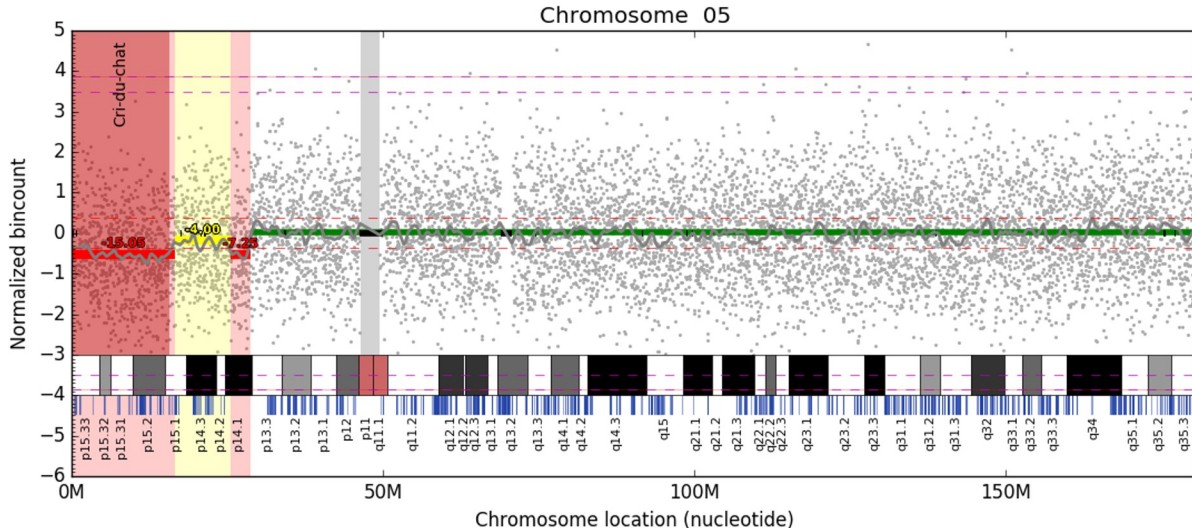

**Fig 1. Visualization of segment significance.** Bin read counts are depicted as gray dots, segments and their significance as colored horizontal lines. Here, we can see an approximately 16.2M long fetal deletion covering the whole Cri-Du-Chat region and a small deletion in the p14.1 region, both depicted in red. However, a possibly fetal deletion depicted by yellow between them may suggest that we are dealing with only one long deletion. Light grey vertical band depicts an unmappable region around the centromere, black bands signify bins that did not pass filtration for healthy characteristics (see Normalisation and filtering in Materials and methods section) and are excluded from the analysis. Light red band emphasizes the detected deletion and pathogenic region. The approximated Z-score of the deletion is displayed over the reported segments. Estimated level of aberration based on the fetal fraction of this sample (12.6%) is visualized as a red dashed line for fetal aberration and magenta dashed line for maternal aberration. This is the one of the very few real NIPT samples with a microaberration available to us.

## Normalization and filtering

To obtain bin counts adjusted for intra- and inter- genomic differences we normalized and filtered the bins by a four-step procedure:

1. LOESS-based GC correction for 20kb bins similar to the one described by Alkan et al. [18]

2. PCA normalization to remove higher-order population artifacts on autosomal chromosomes (similar to [9])

3. filtering of bins with low mean and high variance in bin count

4. subtracting a per-bin mean read count to obtain data normalized around zero

The data used for training both PCA normalization and per-bin mean normalization is available at https://github.com/marcelTBI/CNV_data as bin counts for individual samples (20kb bin size). We used 449 samples for training of PCA normalization (at least 10M reads per sample) and 1238 samples for mean normalization (at least 7M reads per sample). All training samples were collected as standard NIPT samples and checked to be genetically healthy and singleton pregnancies (we include both female and male fetus samples, since we use PCA normalization on autosomal chromosomes only).

In PCA normalization the loess normalized sample bin count matrix was transformed into principal space and first 15 principal components were stored. The first principal components represent common noise in euploid samples [9], so to normalize a sample, we have projected the bin count of a sample to all but the first 15 principal components and thus normalized bin counts to exclude this bias.

Then, only bins that have healthy characteristics were retained–not very low mean and not very high variance. If the read coverage would be uniform, there would be around 7 reads per bin when the sample is normalized to 1M reads. We filtered out bins that had mean read count less than 3.0 (less than half of the ideal mean) or variance higher than 1.5. These numbers were selected manually to ensure that problematic bins would be filtered out, but we still keep around 88% of bins. Note, that this filtering drops all unmappable regions of the genome (centromeres, etc.).

Finally, a per-bin mean read count was subtracted to obtain data normalized around zero. The autosomal chromosomes training samples were firstly rescaled to contain the same number of reads. Since the fetal fraction and fetus gender affects bins of the sex chromosomes, these were rescaled independently of autosomal chromosomes.

## Segment identification and CNV calling

We used a circular binary segmentation (CBS) algorithm provided by the R package "DNA-copy"[19] to identify same-coverage segments. CBS partitions a chromosome into regions with equal copy numbers, therefore it can detect the change point quite precisely. However, this algorithm overly partitions a chromosome. Thus, we used a simple rule to determine the significance of a segment. In an ideal case, the deletion/duplication of fetal chromosome will mean a decrease/increase by a factor $mb^*ff/2$, where $ff$ is the fetal fraction and $mb$ is mean bin count. Since this is a crude simplification and there are uncertainties in the fetal fraction estimation [17,20], and the non-uniform distribution of fetal fraction across the genome [17], we mark as significant all segments that overstep 75% of this theoretical increase/decrease. This percentage can be varied and represents a trade-off between sensitivity and specificity of prediction. The segment was categorized as maternal aberration if it overstepped 75% of maternal level $mb^*(1-ff)/2$. When the fetal fraction is close to 50%, we cannot determine if we are dealing with the maternal or fetal aberration, but these fetal fractions are extremely rare. Furthermore,

this approach is theoretically able to distinguish between maternal and fetomaternal variants, but the distinction is very unreliable due to focus on fetal variants. Thus, in further text we label both fetomaternal and maternal aberrations as simply "maternal". The minimal length of segment categorized as significant was set to 200,000 bases for maternal and 600,000 bases for fetal detections.

The presented method needs to know at least an estimate of the fetal fraction to be able to properly categorize the segments. However, when the fetal fraction is unknown, a small value (5%) can be used to categorize all found deviations from normal as significant at the cost of increase in false positives. We found the 5% fetal fraction as the lowest to reliably detect at least long aberrations, CNV detection of samples with lower fetal fraction is thus not recommended.

## Preparation of control DNA sample mixes

The control DNA sample mixes were prepared as a mixup of healthy female plasma DNA and affected male DNA with confirmed microdeletion syndrome with different ratios to simulate different fetal fraction. Genomic DNA from clinically affected male probands with confirmed microdeletion syndrome were acquired from Coriell Repository biobank (Table A in S1 Text). Additional two anonymized male DiGeorge samples were donated from the Clinic of Genetics at the University Hospital in Bratislava. In addition, genomic male DNA was fragmented with dsDNA shearase according to manufacturer's protocol in order to reach fragments less than 500bp of size. Plasma DNA was received from non-pregnant female volunteers to use as the "maternal" part of the artificial mixup. Sequencing libraries were then prepared and quantified both from non-pregnant plasma and sheared genomic DNA individually, with a PCR-free modification of our previously described method [21]. Libraries were prepared using the Tru-Seq Nano DNA kit (Illumina).

Affected male genomic libraries were then mixed with female healthy plasma libraries to create artificial libraries aiming for different fetal fractions between 5% and 20%. Massively parallel sequencing on NextSeq (Illumina) platform was then performed targeted to 20M uniquely mapped pair-end reads (2x35bp) per sample. This number of reads (instead of more often used 10M) was selected to improve sensitivity, while keeping sequencing costs reasonable. The fetal fraction was measured according to reads mapped on Y chromosomes. Fig 2 and Table A in S1 Text summarize the control DNA sample mixes.

## Results

### Identification of critical regions of chromosomal microdeletion associated syndromes

For the five selected syndromes we identified 570 records corresponding to samples with pathogenic phenotypes and 245 records for benign phenotypes in the ISCA database. These records were used for determination of critical regions. For further analyses, we filtered out outliers and kept 533 regions corresponding to the five selected syndromes. Coverage plots, results of determination of sizes, and positions of all microdeletions associated with syndromes of interest are summarized in Table 2 and visualized in Fig 3. Comparison of studied syndromes in ISCA and DECIPHER databases [22] at the time of our study is available in Table B in S1 Text.

### Analyses of *in-silico* prepared artificial data

To test the sensitivity of detection, we tested the algorithm on 200 different NIPT data sets and on 533 pathogenic regions from the ISCA database. In the first analysis, we kept the read

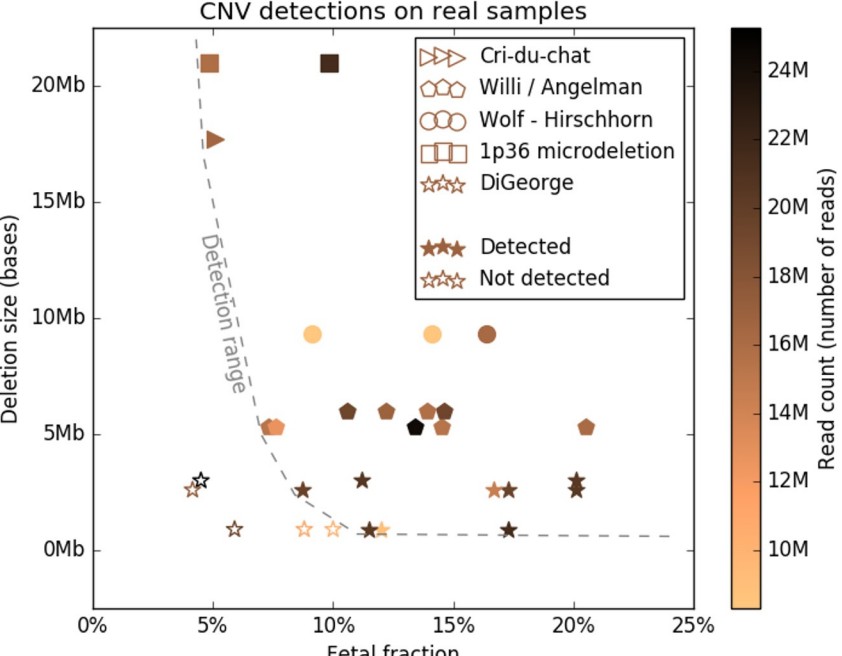

**Fig 2. Control samples and their detection accuracy.** Y axis shows the deletion sizes, X axis the fetal fraction, the color of the point determines its read count. Different syndromes are plotted with different shapes. Detection range was plotted based on all control sample mixups. More detailed data is in the Table A in S1 Text.

count fixed to 20 million (20M) and the fetal fraction varied from 5% to 20% with a step of 2.5%. The second analysis fixed the fetal fraction to 10% and varied the read count from 5M to 20M with a step of 1M.

## Detection accuracy for different fetal fractions

Fetal fraction and size of the deletion were previously reported to be the most crucial factors in detection of chromosomal microdeletions [9]. Sensitivity and specificity calculations for different fetal fractions and sizes of microdeletions at fixed read count to 20M reads per sample were performed (Figs 4 and 5). Data for particular syndromes at 10M reads can be found in the S1 Text.

Out of the 746,200 (533x200x7) carried out simulations, the simulated syndrome was correctly predicted in 571,255 cases (sensitivity = 76.6%). Furthermore, this sensitivity increased to 99.6% if the fetal fraction was at least 10% and the size of the deletion was at least 3Mb. Very poor sensitivity in the 0M-1M range is caused by our strict filtering of detections with small sizes (more info in Materials and Methods), which, as a trade-off, increases specificity.

**Table 2. Size and positions of the pathogenic regions for target syndromes (hg19).**

| Syndrome | Chromosome | Start position | End position |
|---|---|---|---|
| 1p36 deletion | 1 | 564,424 | 21,598,492 |
| Wolf-Hirschhorn | 4 | 85,040 | 2,010,761 |
| Cri-du-chat | 5 | 1 | 15,678,560 |
| Angelman/Prader-Willi | 15 | 22,779,922 | 28,559,437 |
| DiGeorge | 22 | 18,661,724 | 21,505,417 |

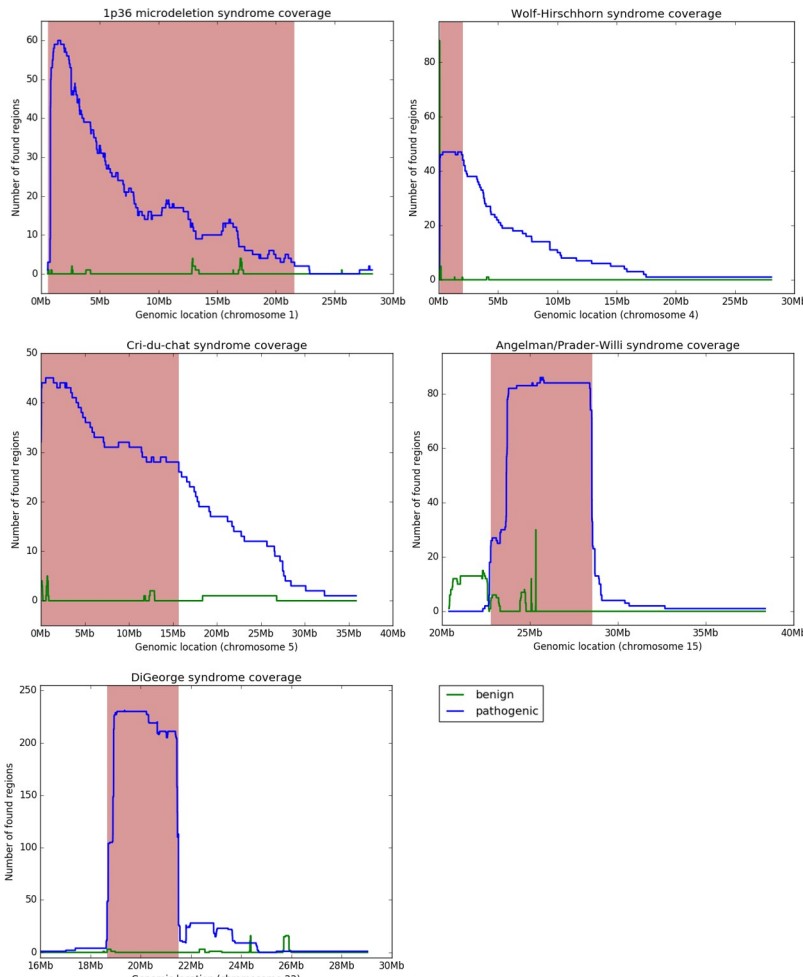

**Fig 3. Coverage plot representing genomic positions of critical regions of five the most common microdeletion syndromes.** Coverage of pathogenic and likely pathogenic deletions is denoted by blue, while coverage of benign and likely benign microdeletions is in green. The critical region is visualized in red.

In Figs A-F in S1 Text, we report sensitivity for a case, when critical regions are enlarged by 2Mb on both sides. However, there is no significant increase of sensitivity.

## Detection accuracy for different read counts

Next analysis focused on estimation of sensitivity of microdeletion detection at different levels of read counts per sample. Fetal fraction was fixed to 10% in this case. Results from calculations for read counts ranging from 5M to 20M of reads per sample are presented in Figs 6 and 7.

Out of the 1,705,600 (533x200x16) carried out simulations, the simulated syndrome was correctly predicted in 937,335 cases (sensitivity = 55.0%). Furthermore, this sensitivity increased to 97.1% if the read count was at least 15M and the size of the deletion was at least 3Mb. Figures for particular syndromes can be found in the S1 Text.

## Validation on control samples

Out of 29 tested DNA samples, created by mixing-up of DNA samples from physiological pregnancies and microdeletion positive control samples, we were able to correctly detect 24

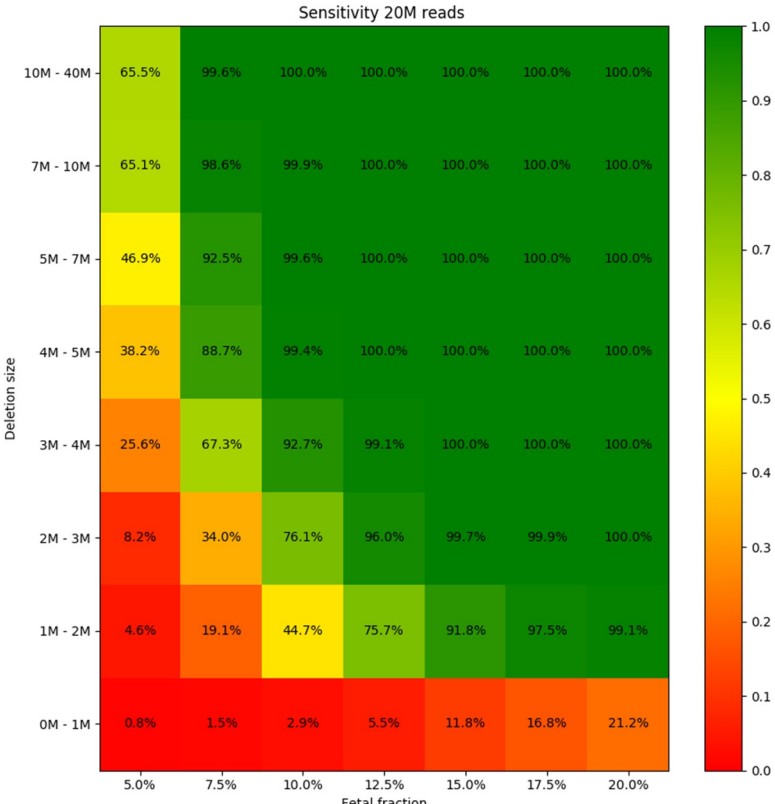

**Fig 4. Sensitivity of the prediction for different fetal fraction and microdeletion size.** Read count was set to 20M in each sample.

(Fig 2 and Table A in S1 Text). The 5 undetected samples had all DiGeorge syndrome, with fetal fraction below 10%, and size of the deletion below 3Mb. Surprisingly, the read count seems to have minimal effect on prediction accuracy, however, we cannot rule out the possibility that limitations of data points skewed this finding.

Our main goal is to evaluate how the accuracy of prediction is dependent on the parameters like fetal fraction and aberration length. Since the real microaberration samples are very scarce, we were forced to use these "simulated" data mixed in the lab from two different samples to be able to, at least partially, control parameters like fetal fraction and aberration length. By using these mixed samples, we introduce a new kind of bias since we are using DNA from two different individuals and also we slightly deviate from standard NIPT protocol. However, the data seem to be consistent with the very few real microaberration samples we encountered in NIPT (data not included).

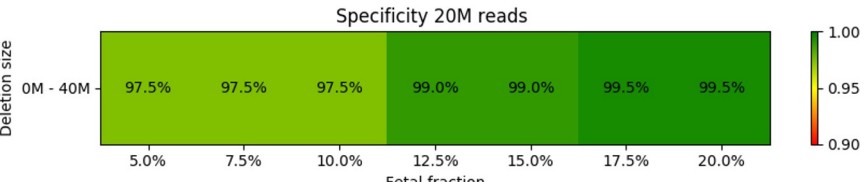

**Fig 5. Specificity of the prediction for different fetal fraction and microdeletion size.** Read count was set to 20M in each sample.

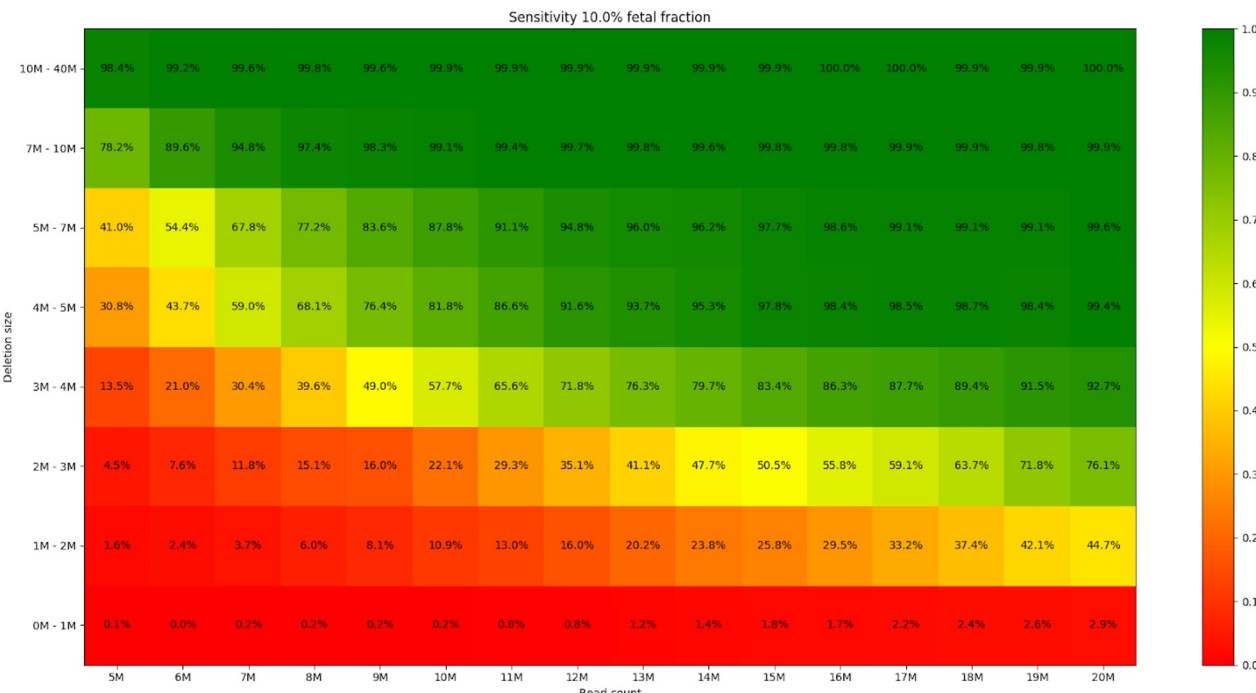

**Fig 6. Sensitivity of the prediction for different read count and microdeletion size.** Fetal fraction was set to 10% for all samples.

## Discussion

Following confirmation of the presence of cffDNA in maternal blood and its use for fetal sex determination [1] first NIPT applications for common aneuploidies were introduced [23] and quickly becoming widely used in prenatal care worldwide [24]. According to the updated statement of the American College of Medical Genetics and Genomics [25] there is strong evidence that NIPT can replace conventional screening for Down, Edwards, and Patau syndrome as it can be performed from 9th gestational week. Nowadays, NIPT is being implemented in public prenatal care and recently has become a standard screening procedure for all pregnant women in the Netherlands [26]. Moreover, the use of whole genome sequencing based tests, allows detection of a wider range of chromosomal aberrations. In line with this, studies using SNP based whole genome scans [6,7] or low coverage whole genome sequencing (0.2x coverage) [9], suggested high sensitivities for detection of the five most frequent microdeletion syndromes. However, these studies had limited means to validate their performance, since real samples with invasively or postpartum confirmed results of NIPT detected microaberrations are available in very limited numbers. Therefore, mostly artificially prepared data or DNA sample mix-ups are generally used for proof of principle pilot studies. Both scenarios were tested in our study yielding comparable results. However, it should be noted that some characteristics of NIPT samples, such as fragmentation patterns, are not realistically represented in

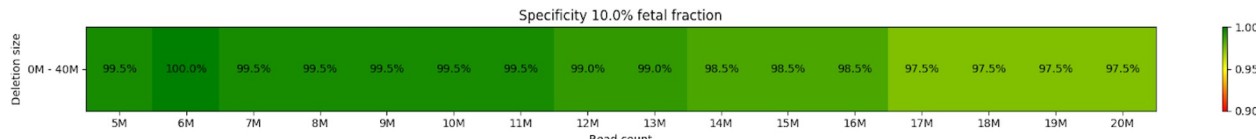

**Fig 7. Specificity of the prediction for different read count and microdeletion size.** Fetal fraction was set to 10% for all samples.

sample mixups of *in-silico* samples [27]. We evaluated performance of our algorithm for detection of microdeletions using low uniquely mapped read count at 10M that is currently considered as standard for reliable detection of Down, Edwards, and Patau syndromes. The motivation for our effort was to define limitations when such low read counts are used. Use of lower read counts is one of key means to reduce substantial sequencing analysis costs per sample. When required read counts drop to a reasonable level around 10M reads, use of middle throughput massively parallel sequencing platforms, such as Illumina NextSeq 500/550, becomes an option, creating thus a potential to a wider adoption of testing across the world.

## Critical region determination

Similarly to our study, Zhao et al. [9] used data from DECIPHER database [22] to define a critical region specific for each of the syndromes. Instead, we used the ISCA database to manually identify the critical region for every syndrome. The main advantage of our approach is the possibility to define critical regions of microdeletion syndromes more specifically. For example DiGeorge syndrome was previously described as 22q11.21 deletion [28], but the real critical region is not the whole band, but only its 3Mb middle part. Our study brings also new information that could be used in the further specification of the size as well as localization of the tested microdeletions as both these parameters were found to be between the four most critical ones [9,29], since there are only very few pathologic detections that overlap the critical regions both from ISCA and DECIPHER databases (Table B in S1 Text).

## Critical parameters for detection of microdeletions

As fetal fraction and deletion size were found to be the most critical parameters in NIPT, different combinations of them were tested. Based on a real range of fetal fraction in routine NIPT testing [6,9], we tested fetal fractions from 5% to 20%. Fetal fractions lower than 5% are problematic due to increased number of false negative detections. For 200 different samples, 533 simulated cases were evaluated. Our approach achieved accuracy of 79.3% for 10% fetal fraction with 20M read count, which further increased to 98.4% if we searched only for deletions longer than 3Mb.

To support our *in-silico* findings, we designed an artificial laboratory sample evaluation test. We used artificial mixtures for all studied syndromes using control DNA samples with precise information about the microdeletion size and position. The only undetected samples were those with DiGeorge syndrome microdeletion shorter than 3Mb and, simultaneously, with fetal fraction lower than 10% (Fig 7). These results are in accordance with the in-silico simulated data.

## Read count importance

Later on, as we tested the influence of read count, the fetal fraction was fixed to 10%, the percentage corresponding to average fetal fraction in pregnant women in the most relevant weeks of pregnancy for NIPT (between 10th and 13th week of pregnancy) [30]. We concluded that the influence of the read count is significant, and the increase of prediction accuracy does not stop at 10M reads (Fig 6), (Fig J in S1 Text). Higher number of reads allows for further increasing of accuracy. Based on our results, we recommend to use approximately 16M-17M (appr. 0.35x genome coverage for 2x35bp reads) reads for analyses, due to fact that the detection rate reaches a plateau for 10% fetal fraction and ≥ 3Mb deletion size around this point (Fig 6). Using even more reads could be beneficial, especially for small deletions and low fetal fractions, but it does not add to the prediction accuracy in most of the tested cases, thus unnecessarily raising the costs of the analysis. On the other hand, if the fetal fraction is higher

(>12.5%) and we focus only on moderate deletion sizes (>3Mb), even coverage as low as 10M reads was shown to be sufficient, suggesting that this test can (and should) be included as part of a basic NIPT even with low coverage. However, it should be mentioned, that deletions causing DiGeorge syndrome (the most frequent syndrome from our list) are usually shorter than 3Mb (Fig 7). Moreover, the genomic location for this syndrome contains a small unmappable region in the middle, which further decreases prediction accuracy. For this syndrome, we recommend at least 20M reads to be used.

## Applicability in routine practice

To test the applicability of *in-silico* results in clinical practice, we have tested control mix-up samples, where results are concordant to those from *in-silico* evaluation (all undetected samples have fetal fraction less than 10% and variation size shorter than 3Mb). As a further proof of the applicability of our approach, we have already detected two microdeletions causing Cri-Du-Chat syndrome and one DiGeorge microdeletion in samples from routine NIPT, where this test is implemented. These findings were subsequently confirmed by conventional methods on samples from amniocentesis.

We should note here, that in the real case scenario, the sensitivity would be slightly lower, since in the experiments we were not dealing with various more rare scenarios like for instance mosaicism. Fetal mosaicism will most likely cause a negative call, since the "effective" fetal fraction for the aberration will be under detection accuracy threshold. On the other hand, maternal mosaicism and fetal aberration can be indistinguishable from the read counts only, which will result in a false positive call. These latter two cases can be distinguished based on the input fetal fraction—if we see an aberration with level that is too far from both expected fetal and maternal levels, we can classify it as "possibly maternal mosaic". However, there are more (unlikely) alternatives—double deletion, multiple duplication, wrongly estimated fetal fraction, etc.

Although we present results for a non-public tool, we believe that the general trends of sensitivity based on fetal fraction, variation length, and read count are transferable to any of the available CNV detection tools, since they all share a common methodology.

## Conclusions

Our results suggest that it is possible to incorporate microaberration detection into whole-genome based NIPT as part of the offered screening/diagnostics procedure, with no or only slight increase in read depths. On the other hand, this is highly dependent on the specific parameters of the used test as well as on the aims of testing. The test has excellent accuracy, when the fetal fraction is above 10% and variation length is above 3Mb, thus becoming a potential diagnostic tool, when these requirements are met. But even when these requirements are not met, the test can detect a significant number of variants, becoming a valuable screening tool in those cases.

However, final decisions on the use and evaluation of the test results, together with specific test parameters, should be a compromise between the cost of the test, objective for testing and the required sensitivity and specificity. Limitations of this approach should always be kept in mind, while professional judgement of a skilled and properly trained evaluator is still in place. Moreover, using this approach, it is possible to distinguish between mother and fetus derived microdeletions, which is based on expected gain or loss of read counts per bin according to determined fetal fraction. This distinction is available for almost the whole range of fetal fractions observed in clinical practice, except for data with fetal fraction around 50%. These would be indistinguishable due to the similar ratio of maternal and fetal DNA. The expected values can be seen on Fig 1 as dashed lines.

## Supporting information

**S1 Text.**
(PDF)

## Acknowledgments

We would like to thank Dr. Iveta Mlkva (Clinic of Genetics at the University Hospital in Bratislava) for kindly providing the control samples with DiGeorge syndrome and for helpful discussions.

## Author Contributions

**Conceptualization:** Jaroslav Budis.

**Data curation:** Marcel Kucharik, Andrej Gnip.

**Formal analysis:** Andrej Gnip.

**Funding acquisition:** Tomas Szemes.

**Investigation:** Marcel Kucharik.

**Methodology:** Marcel Kucharik.

**Resources:** Michaela Hyblova, Maria Harsanyova, Gabriel Minarik.

**Software:** Marcel Kucharik.

**Supervision:** Tomas Szemes.

**Validation:** Michaela Hyblova.

**Visualization:** Marcel Kucharik.

**Writing – original draft:** Marcel Kucharik, Andrej Gnip, Michaela Hyblova, Jaroslav Budis, Gabriel Minarik.

**Writing – review & editing:** Marcel Kucharik, Lucia Strieskova, Ondrej Pös, Zuzana Kubiritova, Jan Radvanszky.

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
