## [Decision Letter · Decision Letter 0]

16 Apr 2020

PONE-D-20-07674

Non-invasive prenatal testing (NIPT) by low coverage genomic sequencing: Detection limits of screened chromosomal microdeletions

PLOS ONE

Dear Kubiritova,

Thank you for submitting your manuscript to PLOS ONE. After careful consideration, we feel that it has merit but does not fully meet PLOS ONE’s publication criteria as it currently stands. Therefore, we invite you to submit a revised version of the manuscript that addresses the points raised during the review process.

We would appreciate receiving your revised manuscript by May 31 2020 11:59PM. To enhance the reproducibility of your results, we recommend that if applicable you deposit your laboratory protocols in protocols.io, where a protocol can be assigned its own identifier (DOI) such that it can be cited independently in the future. For instructions see: http://journals.plos.org/plosone/s/submission-guidelines#loc-laboratory-protocols

We look forward to receiving your revised manuscript.

Kind regards,

Kelvin Yuen Kwong Chan, Ph.D.

Academic Editor

PLOS ONE

Journal Requirements:

"I have read the journal's policy and the authors of this manuscript have the following competing interests:We declare potential competing financial interest in the form of employee contracts (see affiliations for each author) with Geneton Ltd. that participated in the development of a commercial NIPT test in Slovakia. On the other hand, Geneton Ltd. is not a provider of this commercial test, but still continues to do basic and applied research in the field of NIPT. Gnip A, Minarik G and Hyblova M are employees of Medirex Inc./TrisomyTest Ltd. (the commercial providers of NIPT testing in Slovakia), their participation in the study was, however, limited to the routine NIPT testing that generated the genomic results reused in our study. The other authors declare no possible competing interests."

We note that one or more of the authors are employed by a commercial company: Geneton Ltd., Medirex Inc., Trisomy Test Ltd.

4. Your ethics statement must appear in the Methods section of your manuscript. If your ethics statement is written in any section besides the Methods, please move it to the Methods section and delete it from any other section. Please also ensure that your ethics statement is included in your manuscript, as the ethics section of your online submission will not be published alongside your manuscript.

Reviewers' comments:

Reviewer's Responses to Questions

**Comments to the Author**

1. Is the manuscript technically sound, and do the data support the conclusions?

Reviewer #1: Partly

2. Has the statistical analysis been performed appropriately and rigorously? 

Reviewer #1: I Don't Know

3. Have the authors made all data underlying the findings in their manuscript fully available?

Reviewer #1: No

4. Is the manuscript presented in an intelligible fashion and written in standard English?

Reviewer #1: Yes

5. Review Comments to the Author

Reviewer #1: In the paper “Non-invasive prenatal testing (NIPT) by low coverage genomic sequencing: Detection limits of screened chromosomal microdeletions” the authors investigate the feasibility of using cell-free fetal DNA obtained for non-invasive prenatal testing to screen the fetal genome for five clinically relevant microdeletion syndromes using whole genome sequencing based methods.

Currently, NIPT is predominantly focused on the detection of fetal trisomies of chromosomes 13, 18, 21 and the sex chromosomes. Although specific targeted microdeletion tests are offered, they are normally not observed with WGS based methods. For WGS based methods it is not well established what the detection limits and accuracy measures for these syndromes are.

The investigated microdeletion syndromes are sporadic and mostly caused by de-novo deletions that vary in size from ~1.5MB to ~20MB. Boundaries for the actual ‘critical regions’ (the regions that supposedly cause the actual pathogenicity) are often not clearly defined and are mostly smaller than the deletions that are typically observed. Due to the rarity of these syndromes, large scale validation sets are not available. These points complicate the accurate formulation of the performance measures for the detection of these syndromes from NIPT data.

This paper first addresses the problem of defining accurate boundaries for deletions that would cause these syndromes. Then, it uses an ‘in silico’ and a lab-based simulation experiment to determine the sensitivity and precision of detecting deletions that intersect these regions.

In short; the conclusion of this paper is that most syndromes (deletions) can be readily detected from typical NIPT (given a fetal fraction of at least 10% and approximately 20M sequencing reads). The most common DiGeorge syndrome however, is the first to be missed when the fetal fraction becomes too low (<10%).

The paper poses important research questions and answers them sufficiently given the boundaries of the experimental setup.

general remarks:

1) It should be made more clear that the ‘lab mix experiment’ is also a simulated dataset (be it simulated in the lab). No actual fetal micro-deletion pregnancy data is presented. Please reflect on the complexities that will arise when dealing with real microdeletion NIPT data. For instance, what can we expect from the non-uniform distribution of fetal cfDNA and fetal/maternal mosaicism of certain microdeletions, and the huge variation in fetal fraction between samples.

2) Although a complete benchmark is not necessary, the paper would benefit from a comparison of other available tools.

3) The scripts/code does not seem to be publicly available and the methods section alone does not provide enough details on the use of the parameters in order to reproduce these findings accurately. For example, the use of ‘a few steps’ and ‘in-house rules’ (line 127-128), but also ‘bin counts corresponding to these 15 first principal components were removed’, need more in-depth explanation of what is actually happening here to be able to make this reproduceable. Finally, the fact that the segmentation rule uses the exact simulated fetal fraction, seems to be a bit of a fitting procedure. The effect of correctly specifying this parameter on the presented results and the applicability on real NIPT data are unclear to me, and should be addressed. We recommend that you make the scripts available for other users to benefit.

Remarks on the methods section:

Determination of critical/pathological boundaries

To determine the pathological boundaries for each syndrome, the paper describes a somewhat ad-hoc manner in which the most common intersection of deletions in patients with the same syndrome is described, whereas for most of these common syndromes the most common boundaries are thought to be well known. Figures 3 and 4 were more helpful and convincing than the description of filtering criteria. Figures for the other three syndromes are supplied supplementary. Please replace them all by a single Figure in the main text showing the coverage for all five syndromes, as it clearly shows the complexities involved in determining the pathological boundaries, especially for the telomeric microdeletions.

Preparation of artificial NIPT data sets

Based on real NIPT sequencing data, read counts in all bins (of size 20kb) that fall within the pathological boundaries are multiplied by 1-(ff/2) to simulate a deletion. This way, fetal fraction, number of sequenced reads and deletion sizes can be varied in order to test detection accuracy. Although this might be a correct way to simulate the problem ‘in silico’, it should be mentioned in the text that this approach assumes that the fetal fraction is uniformly distributed across the bins, which we know is not the case. Also, a note on mosaicism, might be in place.

Identification of microaberrations

Circular binary segmentation on normalized bin counts; “Identified segments were evaluated using an in-house rule to determine significance”: this needs more detail. It should be replaced by the statistical test that was performed. Or elaborate on the ‘in-house’ rule.

Normalisation

- Then use first 15 PCs from PCA based on reference set of 341 healthy samples:

o Was this based on cfDNA or normal DNA sequencing from 341 samples? No pregnancies?

o “bin counts corresponding to these 15 first principal components were removed”, it is not clear to me what is actually done then? Are certain bins removed directly? Or are bin counts adjusted according to a multiple regression scheme with the 15 PCs as covariates? Maybe add a reference?

o Also, the manuscript says that these 15 PCs now represent ‘common noise’, but couldn’t it be that this is not noise but actual structure that is caused by e.g. common CNVs? Or are there other technical confounders with respect to the sequencing of the reference set (e.g. different sequencing platforms, read length etc.)? Please explain.

Segment identification and CNV calling

- Summary: “Plain circular binary segmentation; over-partitions the genome; fix to pair CBS with a rule based on fetal fraction; theoretical case, mean bin count changes with respect to the fetal fraction in the following manner: mb*ff/2; use segmentation threshold based on the fraction of the theoretical in/decrease.”

- The use of such a rule depends on knowing the actual fetal fraction. In your experiments this is not a problem, as you know the fetal fraction, however, in practice, the fetal fraction can be predicted, but this comes with a large error. The effect of this parameter on the results presented here should therefore be made clear. This seems a weak spot of the presented method, and mentioning this would help readers to be aware of this.

Other:

- Fig 1, is this real, in silico or DNA sample mix data?

- Fig2 ‘Real samples’, probably better to use ‘control samples’, as this is also done throughout the text

6. PLOS authors have the option to publish the peer review history of their article (what does this mean?). If published, this will include your full peer review and any attached files.

Reviewer #1: Yes: Jasper Linthorst and Erik Sistermans

---

## [Author Response · Author response to Decision Letter 0]

16 Jul 2020

Among attached files there is also ´Response to reviewers´file, where both, reviewer and editor comments are answered.

Here I copy the same text as there is in the attached file labeled ´Response to reviewers´:

At first, we would like to thank the reviewers for their critical revision of our manuscript and for the commentaries and suggestions, which were incorporated in our revised manuscript. Our specific replies are provided below. 

Comments to the academic editor and requirements of the journal:

- We properly checked the manuscript and edited some sections (specified below or see in the 'Revised Manuscript with Track Changes') to meets the PLOS ONE's style requirements. In the title page, we added TrisomyTest Ltd. in the affiliation section, since it was missed and deleted Author contribution´ section as it should be upload during online submission. We also updated our Funding Statement and Competing Interests Statement according to requirements. In the Methods section we included the Ethic Statement and edited “Normalization and filtering” section in the Methods section. We also attached ´Data Availability Statement´ and included URL address https://github.com/marcelTBI/CNV_data, where the data used for training both PCA normalization and per-bin mean normalization is available as bin counts for individual samples (20kb bin size). 

Reviewers' comments:

Reviewer's Responses to Questions

Comments to the Author

1. Is the manuscript technically sound, and do the data support the conclusions?

Reviewer #1: Partly

2. Has the statistical analysis been performed appropriately and rigorously?

Reviewer #1: I Don't Know

- Answer to the first and second point, we added some information in to ´Introduction, Results and Discussion section´ and re-written Material and Method section according your comments.

3. Have the authors made all data underlying the findings in their manuscript fully available?

Reviewer #1: No

- We added data used for training both PCA normalization and per-bin mean normalization available at https://github.com/marcelTBI/CNV_data as bin counts for individual samples (20kb bin size). Raw genomic sequences cannot be made publicly available to ensure privacy of study participants (specified in detail in Data Availability Statement). 

4. Is the manuscript presented in an intelligible fashion and written in standard English?

Reviewer #1: Yes

- Thank you.

5. Review Comments to the Author

Reviewer #1: In the paper “Non-invasive prenatal testing (NIPT) by low coverage genomic sequencing: Detection limits of screened chromosomal microdeletions” the authors investigate the feasibility of using cell-free fetal DNA obtained for non-invasive prenatal testing to screen the fetal genome for five clinically relevant microdeletion syndromes using whole genome sequencing based methods.

Currently, NIPT is predominantly focused on the detection of fetal trisomies of chromosomes 13, 18, 21 and the sex chromosomes. Although specific targeted microdeletion tests are offered, they are normally not observed with WGS based methods. For WGS based methods it is not well established what the detection limits and accuracy measures for these syndromes are.

The investigated microdeletion syndromes are sporadic and mostly caused by de-novo deletions that vary in size from ~1.5MB to ~20MB. Boundaries for the actual ‘critical regions’ (the regions that supposedly cause the actual pathogenicity) are often not clearly defined and are mostly smaller than the deletions that are typically observed. Due to the rarity of these syndromes, large scale validation sets are not available. These points complicate the accurate formulation of the performance measures for the detection of these syndromes from NIPT data.

This paper first addresses the problem of defining accurate boundaries for deletions that would cause these syndromes. Then, it uses an ‘in silico’ and a lab-based simulation experiment to determine the sensitivity and precision of detecting deletions that intersect these regions.

In short; the conclusion of this paper is that most syndromes (deletions) can be readily detected from typical NIPT (given a fetal fraction of at least 10% and approximately 20M sequencing reads). The most common DiGeorge syndrome however, is the first to be missed when the fetal fraction becomes too low (<10%).

The paper poses important research questions and answers them sufficiently given the boundaries of the experimental setup.

Comments to the Reviewers´ comments:

General remarks

1) It should be made more clear that the ‘lab mix experiment’ is also a simulated dataset (be it simulated in the lab). No actual fetal micro-deletion pregnancy data is presented. Please reflect on the complexities that will arise when dealing with real microdeletion NIPT data. For instance, what can we expect from the non-uniform distribution of fetal cfDNA and fetal/maternal mosaicism of certain microdeletions, and the huge variation in fetal fraction between samples. 

- Thank you for a valuable remark. We have added notes on mosaicism, fetal fraction and complexities of real data to discussion and results sections. 

2) Although a complete benchmark is not necessary, the paper would benefit from a comparison of other available tools.

- We have included a short overview of available tools to the Introduction and we are working on a comparison of the accuracy of prediction based on the parameters (fetal fraction, aberration length, …) for a few most notable tools - however this is not yet done and much beyond the scope of the article presented. 

3) The scripts/code does not seem to be publicly available and the methods section alone does not provide enough details on the use of the parameters in order to reproduce these findings accurately. For example, the use of ‘a few steps’ and ‘in-house rules’ (line 127-128), but also ‘bin counts corresponding to these 15 first principal components were removed’, need more in-depth explanation of what is actually happening here to be able to make this reproduceable. Finally, the fact that the segmentation rule uses the exact simulated fetal fraction, seems to be a bit of a fitting procedure. The effect of correctly specifying this parameter on the presented results and the applicability on real NIPT data are unclear to me, and should be addressed. We recommend that you make the scripts available for other users to benefit.

- We have expanded the description of the used normalization, filtering, and segment identification. The effect of the used fetal fraction is also discussed at the end of “Segment identification and CNV calling” section. Unfortunately, the scripts cannot be made publicly available as we state in the “Identification of microaberrations” section. Few numbers were changed to reflect recent changes to our CNV detection procedures (the tool is constantly “in development”), but mostly they are only “cosmetic” changes.

Remarks on the methods section:

Determination of critical/pathological boundaries

To determine the pathological boundaries for each syndrome, the paper describes a somewhat ad-hoc manner in which the most common intersection of deletions in patients with the same syndrome is described, whereas for most of these common syndromes the most common boundaries are thought to be well known. Figures 3 and 4 were more helpful and convincing than the description of filtering criteria. Figures for the other three syndromes are supplied supplementary. Please replace them all by a single Figure in the main text showing the coverage for all five syndromes, as it clearly shows the complexities involved in determining the pathological boundaries, especially for the telomeric microdeletions.

- As was suggested, we have created a single figure including Figures 3 and 4 and Figures A, B and C from supplementary material in the main text as Figure 3. In addition to this, we edited figure numbering in the main text and also in the supplementary material.

Preparation of artificial NIPT data sets

Based on real NIPT sequencing data, read counts in all bins (of size 20kb) that fall within the pathological boundaries are multiplied by 1-(ff/2) to simulate a deletion. This way, fetal fraction, number of sequenced reads and deletion sizes can be varied in order to test detection accuracy. Although this might be a correct way to simulate the problem ‘in silico’, it should be mentioned in the text that this approach assumes that the fetal fraction is uniformly distributed across the bins, which we know is not the case. Also, a note on mosaicism, might be in place.

- We have added a remark on non-constant fetal fraction. Mosaicism is discussed in the Discussion. 

Identification of microaberrations

Circular binary segmentation on normalized bin counts; “Identified segments were evaluated using an in-house rule to determine significance”: this needs more detail. It should be replaced by the statistical test that was performed. Or elaborate on the ‘in-house’ rule.

- We have moved all the details of the segment categorization to “Segment identification and CNV calling” section.

Normalisation

- Then use first 15 PCs from PCA based on reference set of 341 healthy samples:

o Was this based on cfDNA or normal DNA sequencing from 341 samples? No pregnancies?

o “bin counts corresponding to these 15 first principal components were removed”, it is not clear to me what is actually done then? Are certain bins removed directly? Or are bin counts adjusted according to a multiple regression scheme with the 15 PCs as covariates? Maybe add a reference?

o Also, the manuscript says that these 15 PCs now represent ‘common noise’, but couldn’t it be that this is not noise but actual structure that is caused by e.g. common CNVs? Or are there other technical confounders with respect to the sequencing of the reference set (e.g. different sequencing platforms, read length etc.)? Please explain.

- We have rewritten the “Normalization and filtering” section to accomodate all of the questions. PCA normalization was done according to the publication mentioned in the previous section - the reference was added to the Normalisation section, too. 

Segment identification and CNV calling

- Summary: “Plain circular binary segmentation; over-partitions the genome; fix to pair CBS with a rule based on fetal fraction; theoretical case, mean bin count changes with respect to the fetal fraction in the following manner: mb*ff/2; use segmentation threshold based on the fraction of the theoretical in/decrease.”

- The use of such a rule depends on knowing the actual fetal fraction. In your experiments this is not a problem, as you know the fetal fraction, however, in practice, the fetal fraction can be predicted, but this comes with a large error. The effect of this parameter on the results presented here should therefore be made clear. This seems a weak spot of the presented method, and mentioning this would help readers to be aware of this.

- We have added a remark on the fetal fraction input to the end of the section.

Other:

- Fig 1, is this real, in silico or DNA sample mix data?

- This is real data. We added it to sample description.

- Fig 2 ‘Real samples’, probably better to use ‘control samples’, as this is also done throughout the text

- In the main text of the submitted manuscript we used “control samples” in the Fig 2.

6. PLOS authors have the option to publish the peer review history of their article (what does this mean?). If published, this will include your full peer review and any attached files.

Do you want your identity to be public for this peer review? For information about this choice, including consent withdrawal, please see our Privacy Policy.

Reviewer #1: Yes: Jasper Linthorst and Erik Sistermans

- We agree with the possibility to publish the peer review history of our article.

We hope that our changes will strengthen our manuscript and that both, the academic editor and reviewers will be satisfied with these changes or modifications.

Sincerely,

Zuzana Kubiritova

---

## [Decision Letter · Decision Letter 1]

13 Aug 2020

Non-invasive prenatal testing (NIPT) by low coverage genomic sequencing: Detection limits of screened chromosomal microdeletions

PONE-D-20-07674R1

Dear Dr. Kubiritova,

We’re pleased to inform you that your manuscript has been judged scientifically suitable for publication and will be formally accepted for publication once it meets all outstanding technical requirements.

Kind regards,

Kelvin Yuen Kwong Chan, Ph.D.

Academic Editor

PLOS ONE

Additional Editor Comments (optional):

Reviewers' comments:

Reviewer's Responses to Questions

**Comments to the Author**

1. If the authors have adequately addressed your comments raised in a previous round of review and you feel that this manuscript is now acceptable for publication, you may indicate that here to bypass the “Comments to the Author” section, enter your conflict of interest statement in the “Confidential to Editor” section, and submit your "Accept" recommendation.

Reviewer #1: All comments have been addressed

2. Is the manuscript technically sound, and do the data support the conclusions?

Reviewer #1: Yes

3. Has the statistical analysis been performed appropriately and rigorously? 

Reviewer #1: N/A

4. Have the authors made all data underlying the findings in their manuscript fully available?

Reviewer #1: Yes

5. Is the manuscript presented in an intelligible fashion and written in standard English?

Reviewer #1: Yes

6. Review Comments to the Author

Reviewer #1: (No Response)

7. PLOS authors have the option to publish the peer review history of their article (what does this mean?). If published, this will include your full peer review and any attached files.

Reviewer #1: No

---

## [Editor Report · Acceptance letter]

17 Aug 2020

PONE-D-20-07674R1 

Non-invasive prenatal testing (NIPT) by low coverage genomic sequencing: Detection limits of screened chromosomal microdeletions 

Dear Dr. Kubiritova:

I'm pleased to inform you that your manuscript has been deemed suitable for publication in PLOS ONE. Congratulations! Your manuscript is now with our production department. 

Kind regards, 

on behalf of

Dr. Kelvin Yuen Kwong Chan 

Academic Editor

PLOS ONE